# FOXM1: Functional Roles of FOXM1 in Non-Malignant Diseases

**DOI:** 10.3390/biom13050857

**Published:** 2023-05-18

**Authors:** Zhenwang Zhang, Mengxi Li, Tian Sun, Zhengrong Zhang, Chao Liu

**Affiliations:** 1Hubei Key Laboratory of Diabetes and Angiopathy, Xianning Medical College, Hubei University of Science and Technology, Xianning 437100, China; zhewangzhang@hbust.edu.cn (Z.Z.);; 2School of Nuclear Technology and Chemistry & Biology, Hubei University of Science and Technology, Xianning 437100, China; 3Medical Research Institute, Xianning Medical College, Hubei University of Science and Technology, Xianning 437100, China

**Keywords:** fox, FOXM1, diabetes, lung disease, new arthritis, life

## Abstract

Forkhead box (FOX) proteins are a wing-like helix family of transcription factors in the DNA-binding region. By mediating the activation and inhibition of transcription and interactions with all kinds of transcriptional co-regulators (MuvB complexes, STAT3, β-catenin, etc.), they play significant roles in carbohydrate and fat metabolism, biological aging and immune regulation, development, and diseases in mammals. Recent studies have focused on translating these essential findings into clinical applications in order to improve quality of life, investigating areas such as diabetes, inflammation, and pulmonary fibrosis, and increase human lifespan. Early studies have shown that forkhead box M1 (FOXM1) functions as a key gene in pathological processes in multiple diseases by regulating genes related to proliferation, the cell cycle, migration, and apoptosis and genes related to diagnosis, therapy, and injury repair. Although FOXM1 has long been studied in relation to human diseases, its role needs to be elaborated on. FOXM1 expression is involved in the development or repair of multiple diseases, including pulmonary fibrosis, pneumonia, diabetes, liver injury repair, adrenal lesions, vascular diseases, brain diseases, arthritis, myasthenia gravis, and psoriasis. The complex mechanisms involve multiple signaling pathways, such as WNT/β-catenin, STAT3/FOXM1/GLUT1, c-Myc/FOXM1, FOXM1/SIRT4/NF-κB, and FOXM1/SEMA3C/NRP2/Hedgehog. This paper reviews the key roles and functions of FOXM1 in kidney, vascular, lung, brain, bone, heart, skin, and blood vessel diseases to elucidate the role of FOXM1 in the development and progression of human non-malignant diseases and makes suggestions for further research.

## 1. Introduction

Diabetes, lung diseases, new arthritis, and cancer are major public health problems worldwide. Due to the dramatic increase in the prevalence of diabetes, the number of people with diabetes is expected to reach 642 million worldwide by 2040 [1]. People with diabetes are at an increased risk of developing cognitive impairment and dementia, which worsens with age—the risk of dementia is increased by 1.5 to 2.5 times in older people with diabetes [2], severely affecting the quality of life of patients. Therefore, the pathogenesis of diabetes should be further elucidated, and effective treatments should be found.

Diabetes is considered a manageable disease, but the adverse effects of poor long-term glycemic management and lifelong treatment are of significant statistical value and carry quality-of-life costs for people with diabetes [3]. Today, scientists worldwide are actively carrying out studies to identify early signs of diabetes and to find targets for the development of new interventions and for countering the metabolic processes that lead to diabetes. A range of lung diseases, such as chronic obstructive pulmonary disease (COPD), pulmonary fibrosis, and acute lung injury (ALI), in addition to toxic environmental and genetic factors, lead to progressive lung diseases characterized by structural destruction of the lungs, disruption of gas exchange, and breathing failure [3].

The FOX family of proteins is a transcription factor with a wing-helix structure in the DNA-binding domain. The transcription factor FOX, either alone or through interactions with a variety of transcriptional coregulatory factors (MuvB complex, STAT3, β-catenin, SMAD3, etc.), regulates the transcriptional activation or transcriptional inhibition of downstream genes [4,5,6,7,8,9] and plays important roles in mammalian development and disease, lipid metabolism and glucose, biological aging, and immune regulation [10]. All FOX proteins share this unique DNA-binding domain but have very different features and functions. FOX genes control various biological functions and are widely expressed during development and in adulthood. Indeed, FOX transcription factors are essential for the normal regulation, differentiation, maintenance, and/or function of the trophoblast ectoderm, vascular tissue, pancreas, intestine, liver, lung, kidney, prostate, ovary, brain, thyroid, bone, bone and heart muscle, and immune cells [11]. Many human diseases are associated with FOX genetic malfunction, such as cancer development (e.g., FOXA1, FOXA2, FOXC2, FOXD1, FOXE1, FOXM1, FOXO1, and FOXO3) [12]. Mouse FOXM1 transcription factors are involved in the internal bile ducts, and the liver’s blood vessels develop during their morphogenesis [13]. FOXO1 may negatively regulate human fetal islet β-cell differentiation by regulating the crucial transcription factors NGN3 and NKX6-1, and the manipulation of FOXO1 levels may be a valuable tool for improving cellular diabetes treatment strategies [14]. The transcription factor FoxG1 affects sensitivity to inflammation by regulating autophagy [15]. Therefore, we believe that the development of drugs targeting the FOX family is an effective and promising strategy that can be implemented to address human diseases in the future when the roles, functions, and mechanisms of the FOX family in disease are well understood. The FOXM1 gene is located on human chromosome 12p13.3 and has 747 encoded amino acids [16]. It has four common isoforms based on the selective splicing of exons: FOXM1a, FOXM1b, FOXM1c, and FOXM1d. Except for FOXM1a, which lacks transcriptional activity, the other subtypes are active transcription factors [17,18]. Thus far, most studies have focused on the active form of FOXM1b.

FOXM1, as a transcriptional regulator, can directly or indirectly regulate cell differentiation, proliferation, metabolism, and apoptosis and the maintenance of stem cell pluripotency [19] (Figure 1). FOXM1 is involved in cellular physiology and pathology through the direct and indirect activation of the transcription and expression of target genes (CCNB1, PLK1, CCNB2, ECT2, EZH2, GPSM2, KIF18B, KPNA2, PHF19, PSRC1, STK17B, and AURKA, etc.) and interacting with various signaling pathways (Wnt/β-catenin, TGF-β/SMADs, STAT3, etc.) [20,21,22,23,24,25,26,27,28]. The following evidence supports the above view: (I) diabetes is caused by FOXM1 mutation [29]; (II) immune cell recruitment regulated by FOXM1 can regulate wound healing in human diabetes [30]; (III) FOXM1 induces hepatocytes to cause inflammation, injury, and fibrosis [31]; (IV) FOXM1 plays a critical role in the induction of mitochondrial biogenesis for human memory-like CAR-T cells from stem cells [32]; and (V) FOXM1 is involved in disease development through the regulation of disease-induced miRNAs [33]. Prior studies have confirmed that FOXM1 is associated with a variety of diseases.

FOXM1 appears to be associated with a variety of cell proliferation diseases. Since FOXM1 itself is regulated by the cell cycle, diseases that affect cell proliferation may also cause a corresponding change in FOXM1 expression. Due to genetic differences, FOXM1 expression is involved in the development or repair of multiple diseases, including pulmonary fibrosis [34], pneumonia [35], diabetes [29], liver injury repair [36], adrenal lesions [37], vascular diseases [38], brain diseases [39], arthritis [40], myasthenia gravis [33], and psoriasis [41]. FOXM1 is involved in signaling pathways associated with disease development and repair, such as FOXM1, which is a key factor in signaling pathways such as c-Myc/FOXM1 [42], FOXM1/SIRT4/NF-κB [43], WNT/β-catenin [44], and FOXM1/SEMA3C/NRP2/Hedgehog [45].

In particular, FOXM1 is involved not only in human diabetes and its complications but also in the advancement of β-cell proliferation and wound healing [30,46]. Recently, several studies have revealed the critical role of FOXM1 in the development and progression of inflammation through the regulation of gene transcription. For example, FOXM1 is involved in regulating genes and signaling pathways such as CRNDE, NF-kappaB, and JAK1/STAT3 [7,47,48].

This article introduces and reviews the functions and mechanisms of FOXM1 in diabetes and its complications, pulmonary fibrosis, inflammation, and myasthenia gravis, as well as the growth and development of patients, in order to provide an essential basis for clinical treatment and applications and future research.

### Forkhead Box M1 (FOXM1): A Simple Summary

FOXM1 is involved in biological processes such as tissue disease, growth, development, and survival through regulating the transcription of genes related to the cell cycle and cell proliferation (Table 1). MMB and FOXM1 have been reported to bind G2/M genes specifically [21,22,49]. MMB and FOXM1 are known to target CCNB1, CCNB2, and PLK1 [21,22]. In addition, the target genes of MMB-FOXM1 also include ECT2, EZH2, GPSM2, KIF18B, KPNA2, PHF19, PSRC1, and STK17B [25].

FOXM1 plays a major role in regulating the development of lung fibrosis by inducing the expression of the proliferation-related genes CCNB1, CCND1, and PLK1 [50]. The insulin-FOXM1/PLK1/CENP-A signaling pathway is able to mitigate beta cell damage and/or prevent diabetes progression by regulating β-cell cycle processes [46]. The transcription factor STAT3 promotes liver regeneration by regulating FOXM1 expression and activating the expression of the hepatocyte-value-added-related GADD46, E2F, and BRCA1/2, as well as other genes [51,52]. It also promotes the expression of the β-catenin, CyclinD1, and c-Myc factors related to the cell cycle and cell value added by increasing the expression of FOXM1 and decreasing the expression of the cell-cycle-inhibition-related factors p21 and p27, thereby improving renal tubular repair and renal function [53]. When FOXM1 is overexpressed, this enhances cell proliferative capacity and delays the onset of premature senescence [54]. It should be noted that FOXM1 predominantly influences the G2/M transition and hardly affects the G1/S transition [55].

FOXM1 regulates many biological processes, such as DNA damage repair, the cell cycle, cell proliferation, cell differentiation, cell renewal, angiogenesis, dynamic tissue balance, cell migration, cell survival, cell senescence, and β-cell islet secretion [18,56]. The roles of FOXM1 in human diseases such as diabetes and its complications, inflammation, fibrosis, psoriasis, and myasthenia gravis, as well as longevity, are detailed below, as this is a key cellular regulator and transcription factor that is involved in regulating biological processes.

**Table 1 biomolecules-13-00857-t001:** FOXM1 target gene.

Transcription Factor	Regulatory Role	Biological Process	References
CCNB1	Activate	Cell cycle	[21] PMID: 22391450 [23] PMID: 23347430
PLK1	Activate	Cell cycle	[9] PMID: 31244930
CDC25B	Activate	Cell cycle	[9] PMID: 31244930 [23] PMID: 23347430
CDC20	Activate	Cell cycle	[22] PMID: 23109430
FZR1	Activate	Cell cycle	[57] PMID: 32152291 [22] PMID: 23109430
CCNB2	Activate	Cell cycle	[22] PMID: 23109430
CCDC44	Activate	Cell cycle	[22] PMID: 23109430
CDK1	Activate	Cell cycle	[22] PMID: 23109430
CENPF	Activate	Cell cycle	[22] PMID: 23109430 [24] PMID: 26100407
OCT4, SOX2, NANOG	Activate	Self-renewal	[58] PMID: 35931301
UBE2C	Activate	Cell cycle	[22] PMID: 23109430
AURKB	Activate	Cell cycle	[24] PMID: 26100407
KNSTRN	Activate	Cell cycle	[24] PMID: 26100407
CCND1	Activate	Cell cycle	[23] PMID: 23347430
SATB2	Activate	Cell proliferation and cell growth	[59] PMID: 33124191
CCNG2	Activate	Cell cycle	[23] PMID: 23347430
ERβ1	Inhibit	cell growth	[60] PMID: 21763263
MYC	Activate	Cell proliferation	[61] PMID: 32802181
Mxi1-SR	Inhibit	Cell proliferation	[62] PMID: 17452451
ERα	Activate	Cell proliferation	[63] PMID: 20208560
ERβ	Inhibit	Cell proliferation	[64] PMID: 35267428
STAT3	Activate	Cell proliferation	[51] PMID: 23110199
Sp1	Activate	Cell proliferation	[65] PMID: 28258481
CTCF	Activate	Cell growth	[66] PMID: 28862757
P53	Inhibit	Cell cycle regulation	[67] PMID: 23300120
FOXM1	Activate	Cell cycle	[68] PMID:19411834, [69] PMID:25254494, [24] PMID: 26100407
SPDEF	Inhibit Activate	Cell proliferation	[69] PMID: 25254494 [70] PMID: 30076647
Twist1	Activate	Cell proliferation	[71] PMID: 24204899
Gli1	Activate	Cell proliferation	[72] PMID: 12183437
Gli2	Activate	Cell stemness	[73] PMID: 29476172
HSF1	Activate	Cell survival	[74] PMID: 37009226
CENPE	Activate	Proliferation	[75] PMID: 31115500
LXRα	Inhibit	Proliferation	[76] PMID: 23812424

## 2. FOXM1 and Lung Diseases

### 2.1. Pulmonary Fibrosis

FOXM1 has an influential function in the development of pulmonary fibrosis. The expression of FOXM1 was found to be increased in idiopathic pulmonary fibrosis (IPF) patients and bleomycin-induced mouse lung fibroblasts. It was shown that the activation of the PI3KA/PDK1/AKT pathway by mitogenic factors (FGF2 and PDGF) promotes the expression of FOXM1, which, in turn, induces the expression of proliferation and survival-related genes, including CCNB1, CCND1, Survivinin, and PLK1, to promote the formation of lung fibrosis [50,68,77]. Interestingly, prostaglandin E2, an endogenous inhibitor of FOXM1, drives AKT and FOXO3a phosphorylation through the modulation of the EP2/cAMP signaling pathway, thereby inhibiting the process of pulmonary fibrosis in mice [78]. The fibroblast-specific deletion of FOXM1 can protect mice from bleomycin-induced fibrosis. The transgene activation of FOXM1 in alveolar epithelial cells (epiFOXM1-ΔN mice) can directly activate the SNAIL1 promoter, induce EMT, increase the expression of IL-1 β, CCL2, and CXCL5, increase inflammatory mediators, and induce fibroblast proliferation [50]. In contrast, the conditional clearance of FOXM1 from alveolar epithelial cells effectively protected mice from the radiation-induced expression of fibrosis-associated factors. It is well known that the pathogenesis of pulmonary fibrosis is closely linked to the production of inflammatory mediators and the epithelial–mesenchymal transition (EMT). As shown above, FOXM1 plays an important role in the progression of radiation-induced lung fibrosis. The expression of FOXM1 is activated in the lungs following radiation induction. A high expression of FOXM1 in vascular endothelial cells enhanced the expression of radiation-pneumonia- and pulmonary-fibrosis-related factors. It increased the expression of inflammatory EMT-related factors such as IL-1 β, CCL2, CXCL5, SNAIL1, ZEB1, ZEB2, and FOXF1. According to its mechanism, FOXM1 induces the transformation of vascular endothelial cells by regulating TGF-β. Furthermore, the repair of inflammatory lung injuries requires the regulation of endothelial HIF-1 α by FOXM1 [79]. Hypoxia-induced pulmonary artery smooth muscle cell proliferation is controlled by FOXM1 [80]. In conclusion, in radiation-induced pulmonary fibrosis, FOXM1 is mediated by the activation of lung inflammation and expression of EMT-related genes (Figure 2 and Table 2).

The regulation of gene expression to inhibit the progression of pulmonary fibrosis (PF) may be a potential and effective new therapy for PF. According to its mechanism, SERCA2a attenuates PF progression by blocking the STAT3/FOXM1 pathway [28] (Figure 2). For example, SERCA2a was found to be lowly expressed in lung tissues from patients with pulmonary fibrosis and in a bleomycin (BLM)-induced mouse model of pulmonary fibrosis. In the bleomycin-induced pulmonary fibrosis model, a high expression of SERCA2a inhibited pulmonary fibrosis progression and vascular remodeling. In prophylactic and therapeutic regimens, a high expression of SERCA2a reduced right ventricular pressure and hypertrophy. In lung cells, the overexpression of SERCA2a inhibited the TGF-β1-induced proliferation and migration of fibroblasts and the conversion of fibroblasts into myofibroblasts.

IPF is a group of chronic lesions associated with high mortality and difficult treatment. Pulmonary macrophages play an important role in IPF by promoting and inhibiting fibrosis, but the molecular mechanisms that regulate macrophage function during fibrosis are unknown. FOXM1 was not expressed in quiescent lungs but was highly expressed in pulmonary macrophages in IPF patients and fibrotic mice. The inactivation of FOXM1 further activates P38MAPK signaling and reduces the expression of DUSP1, a negative regulator of the P38MAPK pathway. The overexpression of DUSP1 in FOXM1-deficient macrophages inhibited the P38MAPK pathway. DUSP1 is known to be a transcription factor FOXM1 target gene.

In conclusion, in macrophages, FOXM1 can stall fibrosis progression by regulating p38MAPK, as compared to fibrosis in known lung epithelial cells and fibroblasts [81] (Figure 2). A drug therapy with particular benefits for pulmonary fibrosis has not yet been identified [82]. Activated fibroblasts are mainly derived from resident stromal fibroblasts, alveolar type II epithelial cells, and bone-marrow-derived fibroblasts through the synthesis and deposition of extracellular matrix proteins [83]. FOXM1 is a key factor in promoting the progression of pulmonary fibrosis and a key target in blocking the progression of fibrosis in lung tissue or other tissues and organs. FOXM1, as a therapeutic target, integrates mitogen and TGF-β features and offers a dual advantage as a therapeutic target compared to targeted drugs that only target mitogen or TGF-β. The development of drugs targeting FOXM1 may be an effective strategy for preventing the progression of pulmonary fibrosis.

FOXM1 is highly expressed in radiation-induced IPF fibroblasts [84]. FOXM1 is necessary not only for mitogen-induced cell proliferation but also for TGF-β1-induced myofibroblast differentiation and resistance to apoptosis. Studies have shown that interference with FOXM1 in fibroblasts or the administration of FOXM1 inhibitors reduces the expression of bleomycin-induced pulmonary-fibrosis-associated factors [50,85]. In conclusion, FOXM1 has been identified as a critical driver and regulator of lung fibrosis activation due to various factors, and the potential for targeting FOXM1 in the treatment of pulmonary fibrosis has been clarified.

### 2.2. FOXM1 and Pulmonary Hypertension

Pulmonary hypertension (PAH) is a degenerative lesion leading to right ventricular failure (RV). RVX208 (BET bromodomain inhibitors) regulates proliferative, pro-survival, and pro-inflammatory pathways through interaction with FOXM1, a clinically available BET inhibitor. It has been shown that FOXM1 plays a vital role in PAH microvascular endothelial and smooth muscle cells and in PAH rat models [86].

### 2.3. FOXM1 and Bronchopulmonary Dysplasia

FOXM1 is a key mediator in lung development and the treatment of bronchopulmonary dysplasia (BPD). FOXM1 is abnormally highly expressed in the lung macrophages of hyperoxy-exposed mice and the lung tissues of patients with BPD. After hyperoxia, the absence of FOXM1 in the bone marrow cell lineage reduces the number of interstitial macrophages, alveolar formation, and lung function. The exaggerated BP-like phenotype observed in highly oxygen-exposed Cre/FOXM1 (−/−) mice was associated with the increased expression of neutrophil-derived myeloperoxidase, protease 3, and protease. FOXM1 affects lung inflammation in response to hyperoxygen, inhibits neutrophil-derived enzymes, enhances monocyte response, limits alveolar damage, and remodels newborn lungs [87].

## 3. FOXM1 and Liver Diseases

### Liver Regeneration

The liver has a strong regenerative capacity. The capacity for liver regeneration may be closely related to the regulation of cell proliferation by FOXM1 [36]. By blocking the hepatic branch of the vagus nerve and inducing the loss or overexpression of hepatocyte-specific FOXM1, it was confirmed that vagal signaling affected hepatocyte proliferation after PHX through the inhibition or activation of the hepatocyte FOXM1 pathway. It is known that the vagus nerve increases mortality after PHX. At the same time, the supplementation of hepatic FOXM1 reduces mortality after PHX, which shows that this is the key to improving survival after hepatic injury. Notably, macrophages serve as mediators of vagal signaling and acute activation by the FOXM1 pathway in hepatocytes. Acetylcholine secreted by the vagus nerve and IL-6 secreted by macrophages are implicated in the molecular mechanisms underlying vagus–macrophage–hepatocyte transduction [88]. According to the molecular mechanism, STAT3 is phosphorylated through the IL-6 signal pathway, p-STAT3 binds to the FOXM1 gene promoter, and the activated FOXM1 gene promotes hepatocyte regeneration [51]. Recent reports have suggested that vagal signals from the liver can directly activate the FOXM1 pathway in pancreatic islet β-cells, promoting a mechanism of compensatory β-cell proliferation [89,90]. Unlike the rich islets, the vagus nerve is rare in the liver and can only be seen around the portal vein. Thus, vagal-signaling-mediated IL-6 secretion upregulates hepatocyte FOXM1 levels and promotes liver regeneration. This complex, multi-step emergency regenerative signaling amplification mechanism of neuronal, immune, and parenchymal cells and FOXM1 promotes rapid liver regeneration and ensures life support after severe liver injury (Figure 3 and Table 1).

## 4. FOXM1 and Kidney Diseases

### 4.1. FOXM1 and Adrenal Lesion 3A Syndrome

FOXM1 plays a role in preventing DNA damage in prostate cells and in a series of cell death processes under oxidative stress. FOXM1 can improve 3A syndrome by regulating changes in genes related to oxidative stress and antioxidant defense [37].

### 4.2. FOXM1 and Acute Kidney Injury

Kidney disease generally refers to various diseases of the kidney. Acute kidney injury (AKI) is a sudden loss of kidney function due to damage to the renal tubular epithelium. Proper renal tubular rejuvenation is critical for preventing the development of chronic renal disease. FOXM1 plays a role in renal tubule repair after acute renal injury. The transcription factor FOXM1 regulates gene expression levels in the AKI model [91]. The expression of FOXM1 increased after renal ischemia–reperfusion in mice. The inhibition of FOXM1 decreased the expression of proliferative factors and blocked cell proliferation and renal tubular regeneration. Glycogen synthase 3 (GSK3) is an upstream regulator of the transcription factor FOXM1. GSK3 inhibits renal tubular repair by inhibiting FOXM1 [92,93]. The inhibition of GSK3 could significantly increase the expression of FOXM1, improve renal tubular repair and renal function, increase the expression of β-catenin, CyclinD1, and c-Myc, and decrease the expression of the cell cycle inhibitors p21 and p27 [53]. In addition, the FOXM1 transcription factor is induced in the early stage of renal injury, which is necessary for epithelial cell proliferation in vitro and relies on the EGFR/FOXM1 signal pathway to drive proliferation and repair after renal injury [94]. In short, FOXM1 plays an important role in renal tubular regeneration after renal injury.

### 4.3. FOXM1 and Renal Fibrosis

The WNT/β-catenin signaling pathway has a significant impact on regulating renal fibrosis. Several members of the WNT family are affected by the regulation of FOXM1. FOXM1 may exhibit a significant switch in activating the β-catenin pathway and renal fibrosis [26]. The inhibition of FOXM1 ameliorates renal interstitial fibrosis (RIF), lowers the mRNA expression of α-SMA and Snail, enhances E-cadherin expression, inhibits the epithelial-to-mesenchymal transition, reduces type I collagen deposition, and significantly reduces structural damage, inflammatory cell infiltration, and extracellular matrix deposition in the kidneys [95]. Therefore, FOXM1 may be a critical factor in the control of renal fibrosis.

### 4.4. FOXM1 and Diabetic Nephropathy

Diabetic nephropathy is a microvascular complication of diabetes associated with systemic and localized renal inflammation and is the primary cause of end-stage renal disease worldwide [96]. Chronic hyperglycemia and hypertension are the main risk factors for diabetic nephropathy. The expression of FOXM1 in the renal tissues of patients with diabetic nephropathy was low. The overexpression of FOXM1 can improve renal function, alleviate pathological changes, and increase the expression of the podocyte marker Nephin in renal tissues [43]. In vitro, FOXM1 increased high-glucose-induced MPC5 cell activity and decreased NLRP3 inflammasome and caspase1 expression. The transcription factor FOXM1 directly regulates sirtuin4 (SIRT4) gene transcriptional activation. In addition, the downregulation of SIRT4 could block the protective effects of FOXM1 in vivo and in vitro. In summary, FOXM1 inhibits the NF-κB signaling pathway through the transcriptional activation of SIRT4, which inhibits the NLRP3 inflammasome and prevents kidney injury and podocyte death in diabetic nephropathy patients. It should be noted that FOXM1 may also positively regulate the expression of the NF-κB target gene and ultimately induce the apoptosis of lymphoblastoid B-cells [97].

## 5. FOXM1 and the Brain

### 5.1. FOXM1 and Brain Development

Cortical expansion and folding are important processes in brain development and evolution. The high expression of FOXM1 can promote the proliferation of basal progenitor cells and cortical thickening and folding. FOXM1 also regulates the proliferation of neural progenitor cells by regulating the expression of Lin28a. In conclusion, FOXM1 can contribute to evolutionary brain development and cortical expansion by increasing the number of basal progenitor cells in mice, promoting brain enlargement and brain rotation [98].

### 5.2. FOXM1 and Hydrocephalus

Long non-coding RNAs (lncRNAs) are associated with many central nervous system disorders. In hydrocephalus, trans-regulation encompasses several lncRNAs involved in pathways regulated by the transcription factor FOXM1 [39]. The pathophysiology of brain injury caused by hydrocephalus remains unclear.

## 6. FOXM1 and the Heart

### 6.1. FOXM1, Myocardial Hypertrophy, and Myocardial Fibrosis

The human embryonic heart developmental process is complex, consisting of multiple genes and signaling pathways that act in concert and can be precisely regulated according to time and space. FOX family proteins are involved in tissue development processes through the regulation of genes’ spatial and temporal expression. They regulate various biological functions, such as the cell cycle, cell proliferation, cell differentiation, migration, metabolism, and DNA damage. Numerous experimental studies in typical biology have confirmed that FOXA2, FOXC1/C2, FOXH1 and FOXM1, FOXOS, and FOXPS are essential regulators of embryonic and cardiac development and play significant roles in cardiac development [11]. Angiotensin-converting enzymes (ACE and ACE2) are crucial for cardiac function. Cardiac stress increases angiotensin-converting enzyme expression, inhibits angiotensin-converting enzyme two expressions, and increases the net production of angiotensin II, leading to myocardial hypertrophy and myocardial fibrosis. Brahma-related gene-1 (BRG1) and FOXM1 acted together in pathologically stressed cardiac endothelial cells, triggering the conversion of ACE2 into ACE and angiotensin I into angiotensin II, respectively, leading to myocardial hypertrophy. In the mouse heart, cardiac stress activates the expression of BRG1 and FOXM1. BRG1 and FOXM1 were activated to form transcriptional protein complexes at the ACE and ACE2 gene promoters, transcriptionally activating ACE and repressing ACE2, promoting angiotensin II production, and leading to myocardial hypertrophy and fibrosis. Interference with BRG1 and FOXM1 or the chemical inhibition of FOXM1 can abrogate the cardiac-stress-induced shift from ACE2 to ACE and protect the heart from myocardial hypertrophy. In conclusion, in the human hypertrophied heart, the BRG1/FOXM1 complex plays a vital role in the prevention/treatment of heart failure by regulating the angiotensin-converting enzyme/angiotensin-converting enzyme ratio [99].

### 6.2. FOXM1 and Myocardial Infarction

FOXM1 has been identified as a potential key regulator of valproic acid (VPA). The high expression of both FOXM1 and VPA showed a cardiac protective effect. Specifically, both VPA treatment and the overexpression of FOXM1 could inhibit the inflammatory response after MI, thus protecting the heart. However, the inhibition of FOXM1 activity blocked the protective effect of VPA on the heart. In conclusion, VPA mediated CM protection through the upregulation of FOXM1 [100].

Bone-marrow-derived growth factor (Mydgf) alleviates cardiac injury after myocardial infarction (MI) in adult mice. Mydgf was found to improve cardiac function through myocardial regeneration. Further RNA sequencing and functional validation revealed that the c-Myc/FOXM1 pathway mediates mydgf-induced cardiomyocyte expansion and the c-Myc/FOXM1 pathway promotes cardiomyocyte proliferation and cardiac regeneration after cardiac injury in neonatal and adult mice, providing potential targets for reversing cardiac remodeling and heart failure [42].

## 7. FOXM1 and Bone/Skeletal Muscle

### 7.1. FOXM1 and Osteoarthritis

Osteoarthritis (OA) is the most common inflammatory joint disease, characterized by the destruction of articular cartilage. Osteoclasts have a unique capacity for bone destruction and are critical in the dynamic balance between bone remodeling and arthritic bone erosion. FOXM1 drives direct mitochondrial biogenesis [101]. The absence of FOXM1 inhibits the ability of arthritis-associated osteoclastogenic macrophages (AtoMs) to become osteoblasts in vitro and in vivo [40]. The global inhibition of FOXM1 reduces joint bone destruction and inflammation. It is therefore possible that cells other than those of the myeloid lineage influence the inflammation process through FOXM1 and thus indirectly affect bone destruction.

Further studies are needed to improve our understanding of this topic. Furthermore, the FOXM1 inhibitor thiomycin inhibits the production of human osteoclasts and forms a target for the potential treatment of rheumatoid arthritis. Osteoporosis is an age-related disease that can lead to the loss of skeletal muscle. Apoptosis and inflammation are the two main factors contributing to osteoporosis. In vitro, PDA causes apoptosis by inducing the expression of the transcription factor FOXM1 and some pro-apoptotic genes, such as PUMA, Bax, and APAF1, in a dose-dependent manner. The molecular mechanism comprises the segregation of the p-NCOR1 and NCOR1-FOXM1 transcriptional complexes and activation of PUMA by FOXM1-mediated transcriptional initiation. The activated PUMA further triggers downstream apoptotic signaling pathways, including the activation of the Bax, APAF1, and caspase cascades, leading to apoptosis [102].

FOXM1 serves as an essential mediator of the inflammatory response. IL-1β is a major pro-inflammatory cytokine associated with cartilage destruction in OA pathophysiology. FOXM1 was upregulated in an in vitro model of IL-1β-induced OA, and the knockdown of FOXM1 blocked the IL-1β-induced activation of NF-κB in chondrocytes [103]. The transcription factors JAK1/STAT3 and FOXM1 are critical mediators of this OA inflammatory response. The interaction between JAK1/STAT3 and FOXM1 was observed in OA. STAT3 could bind to FOXM1, and the inactivation of STAT3 resulted in the downregulation of FOXM1. In addition, the silencing of FOXM1 inhibited LPS-induced inflammatory cytokine production and suppressed the decrease in cell survival. In conclusion, the interaction between JAK1/STAT3 and FOXM1 may play a vital role in the pathogenesis of OA and suggests that the JAK1/STAT3 pathway may be a potential target for OA therapy [7].

### 7.2. FOXM1 and Rheumatoid Arthritis

Rheumatoid arthritis (RA) is a systemic immune disease. FOXM1 is highly expressed in RA models, and FOXM1 may reduce the expression of RND3, activate the Rho/ROCK pathway, and promote the growth and inflammation of fibroblast-like synovial cells (RA-FLSs) [104]. FOXM1 promotes the growth of fibroblast-like synovial cells (RA-FLSs) through the Wnt/β-catenin signaling pathway [61]. The knockdown of FOXM1 reduces the inflammatory response induced by osteoarthritis [7].

### 7.3. FOXM1 and Myasthenia Gravis

FOXM1 was found to be involved in the regulation of myasthenia gravis, which is a transcription factor regulating myasthenia-gravis-related miRNAs [33].

## 8. FOXM1 and Skin-Related Diseases

### 8.1. FOXM1 and Psoriasis

Psoriasis is a common immune-mediated and hereditary skin disease. FOXM1 is a member of the FOX family that has been found to modulate skin diseases. The effect of FOXM1 on the keratinocyte response to tumor necrosis factor-α (TNF-α) was investigated. The results showed that FOXM1 is highly upregulated in psoriatic skin tissues. FOXM1 contributes to psoriasis by regulating TNF-α-induced cell proliferation, apoptosis, and inflammation in keratinocytes [41].

### 8.2. FOXM1 and Systemic Lupus Erythematosus

Systemic lupus erythematosus (SLE) is an autoimmune disease. SLE produces a variety of pathogenic autoantibodies, which are generally understood to be caused by increased type I interferon (IFN) signaling. B-cell subsets of SLE express type I IFN-stimulating genes. We found that the expression of cell-cycle-related genes, specifically the FOXM1 and FOXM1 target genes, was significantly upregulated in SLE patients. Our findings suggest that SLE may be mediated by the FOXM1 enhancement of the proliferative capacity [105].

## 9. FOXM1 and Blood/Vessel-Related Diseases

### 9.1. FOXM1 and Sepsis

The FOXM1 expression of mRNAs was increased 21-fold in septic blood, suggesting that the pathogenic pathway of sepsis may also be closely related to FOXM1 [106].

### 9.2. FOXM1 and Vascular Disease

The transcription factor FOXM1 is required by smooth muscle cells during the embryonic development of the vasculature and esophagus [107]. FOXM1 is a crucial regulator of dysfunctional SMC cell proliferation and promotes pulmonary vascular remodeling by regulating SMC cell proliferation. In addition, FOXM1 plays vital roles in vascular repair mechanisms by restoring vascular integrity, eliminating inflammation, and promoting repair after inflammatory lung injury. Therefore, the targeting of FOXM1 is a novel strategy for the treatment of idiopathic pulmonary arterial hypertension (PAH) [108,109].

## 10. FOXM1 and Diabetes Mellitus and Its Complications

### 10.1. FOXM1 and Diabetic Foot Ulcers (DFUs)

DFU is a diabetic complication that severely impacts the quality of life of patients and is associated with high mortality. However, the molecular mechanisms underlying the development of DFU are still poorly understood. The inhibition of FOXM1 expression in mouse diabetic models (streptozotocin-induced and db/db) delayed wound healing and decreased neutrophil and macrophage recruitment in diabetic wounds [30] (Figure 4 and Table 2). Diabetes mellitus drives the neutrophil formation of extracellular neutrophils, which may lead to decreased tissue damage and healing capacity. DFU may occur due to a decrease in the number of neutrophils due to FOXM1 deficiency, leading to a disrupted immune response. It was found that FOXM1 regulates reactive oxygen species (ROS) levels in neutrophils, and the inhibition of FOXM1 leads to an increase in ROS, resulting in the formation of DFU. One study showed that TREM1 promotes the recruitment of FOXM1-positive neutrophils, reverses DFU, and promotes wound healing [110] (Figure 3). In addition, some investigators observed collagen deposition, angiogenesis, and human dermal fibroblast (HDF) proliferation and migration, as well as macrophage polarization, following the ectopic expression or knockdown of FOXM1. The results showed that FOXM1 was lowly expressed in the traumatic tissues of DFU rats. The silencing of FOXM1 reversed the promotion of HDF proliferation and migration induced by M2 polarization. Further results showed that FOXM1 accelerated diabetic foot wound healing by regulating elevated SEMA3C expression, upregulating NRP2, and activating the Hedgehog signaling pathway to promote M2 polarization and HDF proliferation [45]. To sum up, the FOXM1 pathway is a new regulatory factor of network formation in the process of diabetic wound healing, which provides a new therapeutic strategy for promoting DFU healing.

### 10.2. The Rolse of FOXM1 in β-Cell Proliferation and Activity

FOXM1 is critical for the progression of the cell cycle in mammalian cells. The capacity for mammalian pancreatic β-cell damage repair or proliferation is closely related to the regulation of cell proliferation by FOXM1. The cell cycle progression of mammalian pancreatic β-cells is focused on the transition from the stationary phase (G0) to the G1 phase. FOXM1 and centromere protein A (CENP-A) are required for chromosome segregation in the M phase of the β-cell cycle and for β-cell proliferation [46]. The insulin-receptor-mediated signaling pathway activates FOXM1 transcriptional activity, and FOXM1 interacts with CDK1/CDK2 to regulate the expression of CENP-A and PLK1. A lack of FOXM1 in pancreatic β-cells leads to impaired β-cell adaptive proliferation in mice in response to pregnancy, acute and chronic insulin resistance, or cellular senescence. Thus, mitotic cell cycle progression, regulated by the FOXM1 pathway, is important in delaying β-cell adaptation and/or preventing progression to diabetes [46] (Figure 5). Mice that were deficient in FOXM1 in the pancreas exhibited glucose tolerance or diabetes and had a 60% decrease in β-cell mass, indicating that the deletion of FOXM1 is detrimental to β-cell function. Furthermore, FOXM1-deficient mice exhibited reduced insulin secretion, whereas the activation of FOXM1 increased β-cell replication, enhanced insulin production, and improved glucose balance, rendering FOXM1 an appealing target for the management of diabetes [111] (Figure 4). FOXM1 is a regulator of cell cycle progression in β-cells and is necessary for β-cell expansion after pregnancy, labor, or partial pancreatectomy, as well as β-cell expansion and glucose level stabilization during pregnancy [112,113]. However, the upregulation of unclipped FOXM1 did not affect stimulated β-cell mass in non-stressed mice or after partial pancreatectomy due to the fact FOXM1 was in a non-activated state. Activated FOXM1 contributes to the recovery of β-cells after injury. HA-FOXM1-NRD transgenic mice showed decreased glycemia and increased β-cell mass in an STZ-induced diabetes model. Compared to rIP-RTTA: Teto-HA-FOXM1 mice, β-cell death was significantly reduced in the rIP-RTTA: Teto-HA-FOXM1 group after seven days of STZ treatment. Further results showed that activated FOXM1 altered gene expression in the extracellular matrix and immune cells, preventing STZ-mediated β-cell death. These studies emphasize that FOXM1 is a critical regulator of pancreatic β-cell survival and cellular proliferation [114,115,116].

### 10.3. FOXM1 Is Implicated in Pathways or Key Regulatory Genes Related to β-Cell Proliferation and Activity

The transcription factor FOXM1 and its target genes are important for nutrition-induced β-cell proliferation. The combined infusion of AG1478 (EGFR inhibitor) and rapamycin (mTOR inhibitor) blocked FOXM1 signaling in 6-month-old rats, which, in turn, inhibited the increase in β-cell proliferation, β-cell mass, and β-cell volume because the nutrition-induced β-cell proliferation involved the activation of EGFR/FOXM1 and mTOR/FOXM1 signals [117,118,119] (Figure 6).

CENP-A and PLK1 are cell cycle M-phase-related proteins regulated by FOXM1, a transcription factor downstream of insulin signal transduction (Figure 7). Therefore, the FOXM1/PLK1/CENP-A signaling pathway is vital in β-cell senescence, diet-induced obesity (DIO), pregnancy, and acute insulin resistance [120]. In addition, further studies have shown that luseogliflzin can promote β-cell proliferation by acting on the FOXM1/PLK1/CENP-A pathway through humoral factors in a non-insulin-/IGF1-receptor-dependent manner [121]. Therefore, the FOXM1/PLK1/CENP-A pathway is the fundamental regulatory axis of glucose signal-induced adaptive β-cell proliferation initiated by glucokinase. FOXM1 has a rare mutation, rs535471991, which significantly reduces the stability, activity, and function of the FOXM1 phosphorylation structural domain. It was shown that FOXM1 plays a critical role in the diagnosis and treatment of maturity-onset diabetes of the young (MODY) through effects such as reducing the rate of misdiagnosis. The design of drugs targeting FOXM1 may provide promising treatments for MODY patients [29].

Mouse insulin promoter (MIP) was combined with the transcription factor FOXM1 gene to construct transgenic mice specifically targeting pancreatic β-cells under the control of MIP. The high expression of FOXM1 in islets promoted the proliferation of β-cells, resulting in a decrease in blood glucose and an increase in pancreatic insulin levels in MIP-FOXM1 mice [122]. In addition, YAP-dependent β-cells need FOXM1 for proliferation. YAP/FOXM1 activation can protect β-cells from apoptosis triggered by various diabetic states. YAP/FOXM1 is considered an effective target axis for improving the remission of functional β-cell disease in patients with diabetes [123]. Under the action of dibutyl phthalate (DBP), the expression of pSTAT1 was high, while FOXM1 was significantly inhibited by pSTAT1. It was found that DBP affects the progression of gestational diabetes by significantly increasing the expression of pSTAT1 and inhibiting FOXM1, decreasing the β-cell survival rate [124]. The integrity of DYRK1A/B activity is the key factor controlling β-cell replication, while FOXM1 is the crucial DYRK1A/B component. The inhibitory effect of DYRK1A/B inhibitor (CC-401) on β-cell growth was studied, providing a reference for understanding the molecular pathway of β-cell growth (Figure 7).

FOXM1 participates in the proliferation of β-cells induced by placental prolactin (PL). FOXM1 was found to be highly expressed in maternal islets during pregnancy, and PL could induce the expression of FOXM1 in cultured islets. Studies have shown that FOXM1 acts downstream of PL and mediates its effect on β-cell proliferation [125]. FOXM1 is necessary for glucose homeostasis and the immense expansion of maternal islet β-cells during pregnancy. In cases of insulin resistance, such as obesity, pancreatic β-cells proliferate to prevent the increase in blood glucose. Liver–brain–pancreatic neurons play an important role in this process. The blockage of this transmission, including vagotomy, can inhibit the activation of the FOXM1 pathway in β-cells induced by obesity and inhibit β-cell expansion, a molecular mechanism of compensatory β-cell proliferation. Inducible β-cell-specific FOXM1 defects also blocked the compensatory proliferation of β-cells. In isolated islet β-cells, carbachol and PACAP-/VIP-dependent FOXM1 synergistically regulate β-cell proliferation. The above studies suggest that the vagus nerve promotes β-cell insulin secretion by releasing multiple neurotransmitters that activate multiple pathways of β-cell value-added factors, thereby rescuing glycemic stability during the development of obesity. This FOXM1 synergistically regulated and neuronal-signaling-mediated mechanism raises the possibility of the regeneration of pancreatic islet β-cells [90]. In pancreatic islet β-cells, FOXM1 is closely associated with obesity. It is known that body mass index is positively associated with FOXM1 expression. Obese adult β-cells can compensate for insulin resistance by promoting β-cell proliferation and improving β-cell mass. This is because the FOXM1 transcriptional program plays a crucial role in this process [115].

In conclusion, FOXM1 deficiency inhibits β-cell proliferation, thus suggesting that FOXM1 plays undeniable roles in islet β-cell proliferation and cell survival. Short-term FOXM1 deficiency has no significant effect on β-cell mass or glucose homeostasis. However, long-term FOXM1 deficiency decreases β-cell mass, hinders insulin secretion, and disrupts glucose homeostasis. Therefore, the decrease in β-cell number is the main reason for impaired insulin secretion caused by long-term FOXM1 deficiency. In adult β-cells, the stable expression of FOXM1 is an essential factor for maintaining a low level of cell proliferation in the case of β-cells, sufficient β-cell mass, and good glucose tolerance. Additionally, the preservation of β-cell FOXM1 activity can prevent β-cell senescence and the impairment of glucose tolerance with age [126].

## 11. FOXM1 and Immune Response to Non-Malignant Diseases

Studies have shown that in DFU models, FOXM1, STAT3, and TNF alpha adjustment factor are significantly suppressed, leading to immune cells (macrophages and neutrophils, monocytes, Langerhans cells) in the DFU environment by inhibiting activation, proliferation, and survival. This, in turn, leads to uncontrolled inflammatory responses and inhibited wound healing. In acute oral wound and acute skin wound model groups, FOXM1, STAT3, and TNF-α regulatory factors are significantly activated. This leads to immune cells’ (macrophages and neutrophils, monocytes, Langerhans cells) activation, proliferation, and survival, thus promoting wound healing [30]. DFUs activate neutrophils to form NETs, leading to tissue damage and impaired healing. According to the main mechanism, FOXM1 regulates the level of ROS in neutrophils, and the decrease in FOXM1 is detected by DFUs to promote the increase in ROS and the formation of NET. The increase in TREM1 promotes FoxM1-positive neutrophil recruitment, reverses the effects of diabetes, and promotes wound healing in the body [110].

The transcription factors JAK1/STAT3 and FOXM1 are key mediators in OA. In OA, the interaction of JAK1/STAT3 and FOXM1 promotes inflammation and cartilage injury, the macrophages and neutrophils are activated, and the immune response is disturbed. The silencing of FOXM1 inhibited the production of LP-induced inflammatory cytokines and decreased cell survival [7].

FOXM1 is a key factor in BPD. Bone marrow cell lineages with FOXM1 deletion in hyperoxic models reduced the number of interstitial macrophages, affected alveolar formation, and impaired lung function. Neutrophil activation was observed in a model of BPD in a study on hyperoxy-exposed Cre/FOXM1(−/−) mice [87].

FOXM1 is believed to be a downstream target of NOTCH and is responsible for the abovementioned metabolic changes and subsequent stem cell memory-like T (iTSCM) differentiation. As with NOTCH-induced CAR iTSCM cells, FOXM1-induced CAR iTSCM cells showed stronger immune response potential compared to conventional CAR T cells [32].

## 12. FOXM1 and Lifespan

The FOX gene is a transcription factor that can drive the expression of other genes and is known to play an essential role in cell proliferation and the lifespan. FOXO3, a specific FOX gene family member, is directly related to the human lifespan. FOXM1, another forkhead box gene, is an essential regulator of oxidative stress response and one of the prominent participants in tumorigenesis. It is a leading object of research on the elderly. FOXM1 is a vital transcription factor protein that regulates the cell cycle. When FOXM1 was removed from mice, the young mice died of heart failure shortly after birth. Previous studies have shown that approximately 60% of the transcriptional changes related to the senescence of human skin fibroblasts depend on the transcription factor FOXM1, and FOXM1 is a crucial proliferation-associated transcription factor that primarily regulates the expression of genes associated with cell cycle G2/M phase progression [54]. In the field of senescence, it is often considered that the loss of cell proliferation is an essential marker of cellular senescence. In healthy older adults and Hutchinson–Guildford premature aging syndrome caused by LMNA gene mutation, FOXM1 decreases with accelerated aging, and the induction of FOXM1 expression can delay the senescence of these cells. Moreover, further studies showed that FOXM1 expression can bypass aging [54,127]. All this evidence suggests that FOXM1 is closely related to aging; thus, could it be used as a therapeutic target? The latest research has shown that the transcription factor FOXM1 can prolong the median lifespan of experimental mice by 28% and the maximum lifespan by 29% and reduce aging-related weight loss in mice [128,129,130].

How might senescence be delayed with FOXM1 in premature senile mice and naturally aging mice? The following findings are relevant to this question: (I) DNA damage and nuclear abnormality decreased, and cell proliferation increased, finally resulting in the epigenetic state of reducing senescence and recovery. The N-terminal deleted form of FOXM1 (FOXM1-dNdK) was found to be overexpressed in HGPS (Hutchinson–Gilford progeria syndrome)-derived fibroblasts, and FOXM1-dNdK provided cell proliferative capacity and delayed the onset of premature senescence [54]. (II) FOXM1 retarded and accelerated aging in the HGPS mouse model [54,128]. FOXM1 not only reduced bone defects and growth retardation in mice but also improved heart function, and the lifespan of these mice was approximately 25% longer than that of untreated premature senile mice. In addition, skin homeostasis, aortic wall thickening, and bone mineral density and volume were improved in the premature senile mice [130]. (III) FOXM1 delayed natural aging by 30% [128]. FOXM1-dNdK was re-used in 3-day and 4-day protocols for animal experiments on naturally aging mice ranging from 8 weeks to 80 weeks of age [128]. The researchers found that the treatment extended the life expectancy of the naturally aging mice by nearly 30% compared with the control group. Histological examinations showed that the truncation of FOXM1 was induced to rejuvenate multiple organs: the aorta, skin, fat, and muscle. It was observed that muscle atrophy decreased, the number of muscle stem cells increased, and muscle strength increased. In addition, it was confirmed that aortic fibrosis and wall thickening decreased, and subcutaneous fat increased. This study also confirms previous results showing that naturally aging mice exhibit downregulated aging biomarkers in the muscle, skin, fat, and kidneys after truncated FOXM1 induction [128,129,130].

The activation of FOXM1 gene expression restores the aging process and mitigates the decline in aging-related factors. FOXM1 can also counteract physiological aging through tissue-specific molecules that regulate inflammatory and metabolic pathways, being positively correlated with the molecular profiles of the currently recognized anti-aging protocols. In conclusion, FOXM1 can delay pathological aging, including premature and natural aging.

## 13. Concluding Remarks

In this review, we presented some interesting findings that may contribute to clinical applications and future research. In addition to the occurrence and development of tumors, FOXM1 plays important regulatory roles in the growth and development of diabetes and occurrence and development of kidney, vascular, lung, brain, bone, heart, skin, and blood vessel diseases, as well as tissue damage repair.

In recent years, FOXM1 has emerged as an important regulator of a variety of pathological processes. The critical role of FOXM1 in cancer confirms its importance for therapy and intervention. However, its roles in other human diseases have not been reviewed in detail. The current data suggest that single or combination therapy targeting FOXM1 may be effective in the treatment of diseases such as pulmonary fibrosis and inflammation. Currently, the roles of FOXM1 in various diseases are being clarified. However, there are no clinical drugs targeting FOXM1. Further research is needed to explore the role of FOXM1 in more forms in the future. However, it is encouraging to note that a growing number of development strategies are accelerating this research process, which gives us hope that the current problems may be overcome. However, the quality and quantity of siFOXM1 delivery to pathological cells are challenging issues. M1-138 has been shown to reduce cell proliferation by binding to FOXM1 and the FOXM1-interacting factor SMAD3, which are peptides developed to target FOXM1 [9]. Few small-molecule drugs that effectively and specifically inhibit FOXM1 are currently available, which has slowed the development of drugs targeting FOXM1 as effective treatments for human disease. We anticipate better diagnosis and treatment of diabetes, congenital diseases, and other developmental disorders in the future. However, the targeting of FOXM1 may be a double-edged sword, as the role of FOXM1 in promoting fetal growth and development, in repairing tissue damage, or in the dynamic balance of energy metabolism also influences disease onset and progression. The challenge for the current drugs targeting FOXM1 lies in the ability to regulate FOXM1 expression and activity under specific pathological conditions, which requires an accurate understanding of FOXM1 regulation and function in health and disease. This article reviewed the roles of FOXM1 in various diseases, particularly chronic progressive diabetes, pulmonary fibrosis, and organismal aging. It described the roles and functions of FOXM1 in heart, brain, vascular, lung, and liver tissue/organ injury and pathology (Figure 8).

This article also reviews the roles of FOXM1 in the treatment and occurrence of lung diseases, liver diseases, kidney diseases, brain-, cardio-, bone-, and skin-related diseases, diabetes, and other diseases as a therapeutic target with broad prospects. In terms of diabetes and its complications, the regulation of FOXM1 can improve insulin resistance and insulin secretion and reduce the occurrence and development of diabetes. In kidney disease, lung disease, heart disease, and brain disease, FOXM1 regulation reduces inflammatory responses and apoptosis, promoting tissue repair and regeneration. In arthritis, FOXM1 regulation may inhibit inflammatory responses and joint destruction, reduce pain, and improve joint function. Therefore, the future prospects of FOXM1 as a therapeutic target are a cause for great optimism.

## Figures and Tables

**Figure 1 biomolecules-13-00857-f001:**
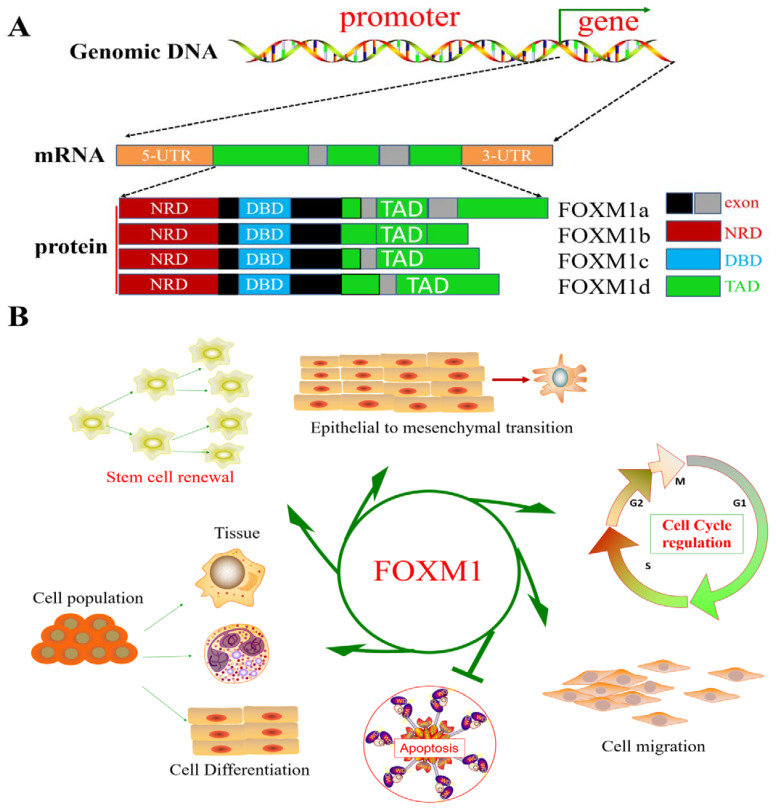
Genomic structure and coding isoforms of the FOXM1 gene and FOXM1 function. (**A**) FOXM1 genome structure and splice isoforms. (**B**) Schematic diagram of FOXM1, which plays a role as a transcriptional regulator of cell differentiation, proliferation, metabolism, and apoptosis and in maintaining stem cell pluripotency.

**Figure 2 biomolecules-13-00857-f002:**
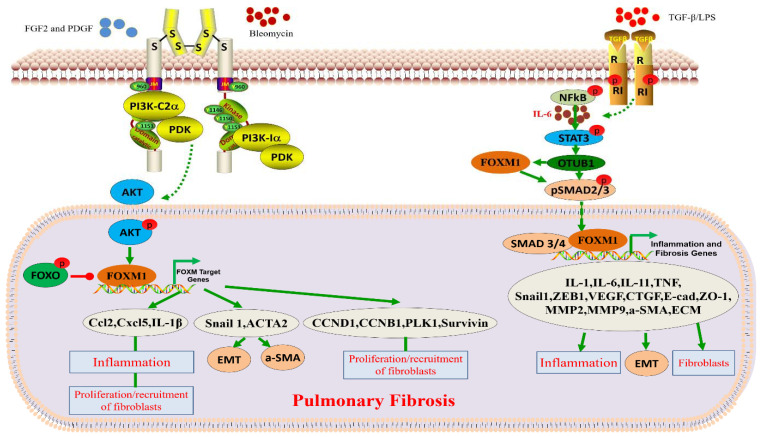
Role and function of FOXM1 in pulmonary fibrosis. Bleomyclin, TGF-β, and/or LPS contribute to the development of pulmonary fibrosis by activating cell receptors and further activating FOXM1-related signaling pathways.

**Figure 3 biomolecules-13-00857-f003:**
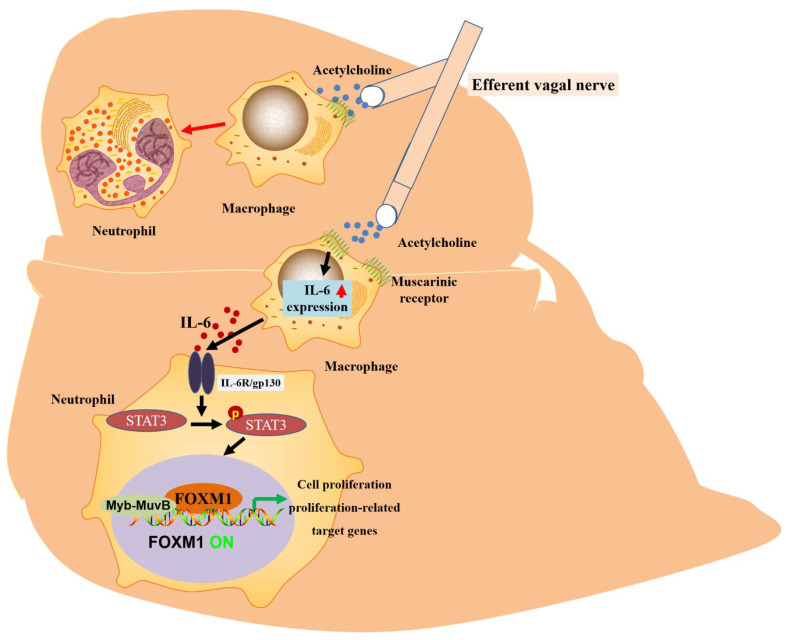
The proposed mechanism of vagus-mediated liver regeneration. Muscarinic signals from the vagus nerve may stimulate IL-6 production by hepatic macrophages, thereby activating the hepatocyte FOXM1 pathway in a paracrine manner, in which resident macrophages act as mediators and regenerative signals from the vagus nerve can be secreted and propagated throughout the remnant liver via IL-6, thereby enhancing compensatory proliferation.

**Figure 4 biomolecules-13-00857-f004:**
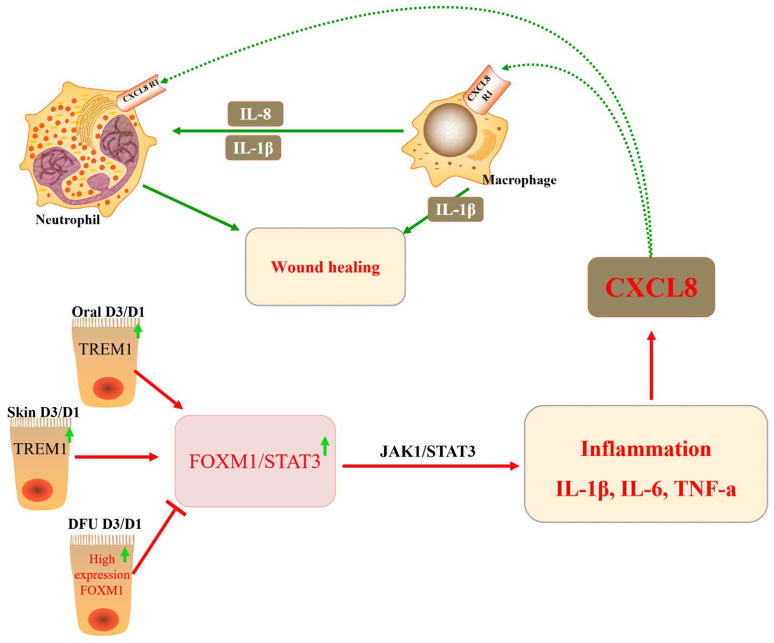
The role of FOXM1 in diabetic foot ulcers (DFUs). The transcription factor FOXM1 regulates reactive ROS levels in neutrophils, leading to inflammatory responses, which, in turn, activate the chemokine CXCL8 and promote wound healing in vivo.

**Figure 5 biomolecules-13-00857-f005:**
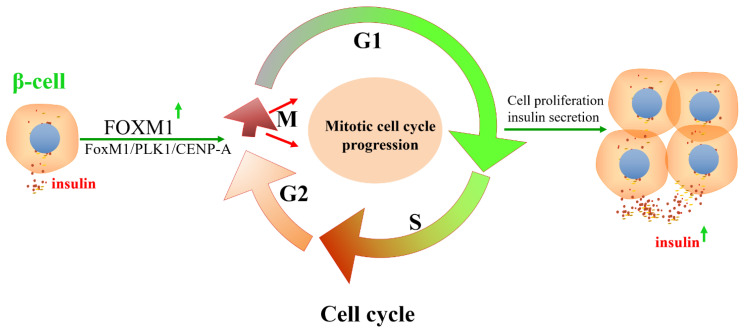
FOXM1/PLK1/CENP-A pathway regulates the cell cycle progression of β-cells and insulin secretion. The insulin signaling pathway promotes the DNA-binding activity of FOXM1 and regulates the expression of CENP-A and Polo-like kinase-1 (PLK1) by regulating cyclin-dependent kinase 1/2. β-cell mitotic cell cycle progression and insulin secretion are regulated by the insulin-FOXM1/PLK1/CENP-A pathway.

**Figure 6 biomolecules-13-00857-f006:**
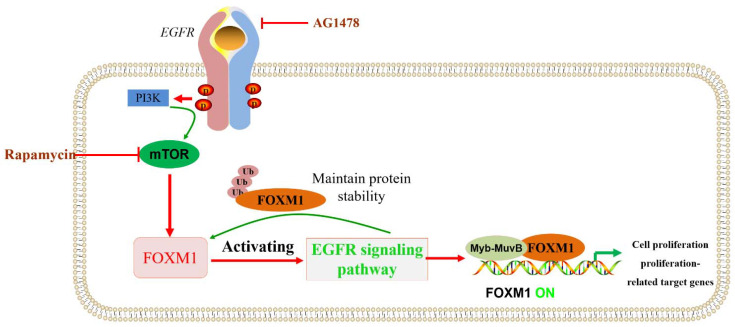
EGFR/mTOR signaling pathways that regulate FOXM1 are involved in β-cell proliferation. PI3K/mTOR and/or EGFR signaling pathways regulate or maintain FOXM1 protein levels, being involved in biological processes such as cell proliferation.

**Figure 7 biomolecules-13-00857-f007:**
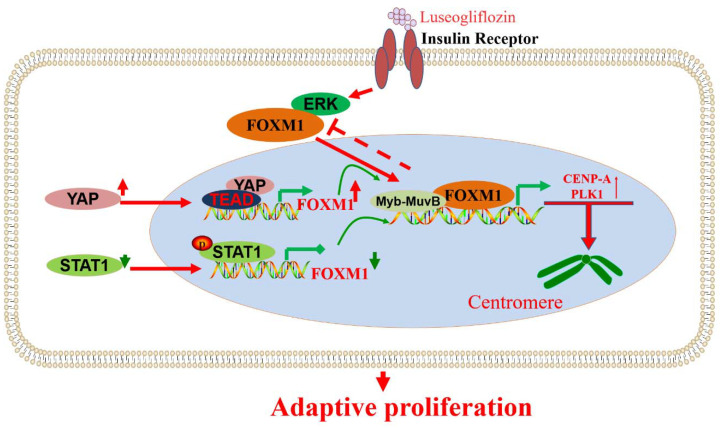
Signaling pathways or key regulatory genes that regulate FOXM1 are involved in β-cell proliferation and activity. YAP, STAT1, ERK, and other factors are involved in the regulation of FOXM1 and further affect the role of FOXM1 in the proliferation of β-cell cells.

**Figure 8 biomolecules-13-00857-f008:**
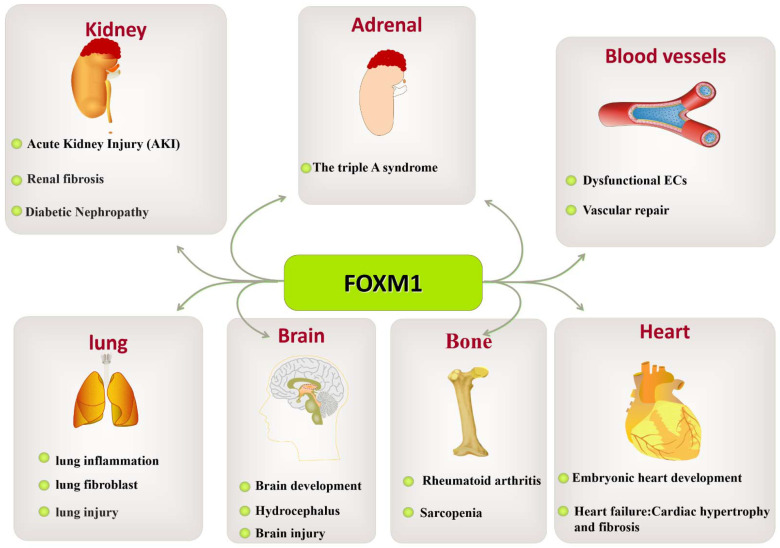
The role of FOXM1 in various organ diseases. The abnormal expression of FOXM1 leads to pathological changes in various tissues and organs, such as kidney-related diseases (triple-A syndrome, AKI, renal fibrosis, diabetic nephropathy, etc.), vascular-related diseases (dysfunctional ECs, vascular repair), lung-related diseases (lung fibroblasts, lung injury), brain-related diseases (brain growth and development, hydrocephalus, brain injury), bone-related diseases (rheumatoid arthritis), and heart-related diseases (embryonic heart development and heart failure: cardiac hypertrophy and fibrosis).

**Table 2 biomolecules-13-00857-t002:** FOXM1 target gene and biological function.

Target Gene	Biological Function
Ccl2	Inflammation
Cxcl5	Inflammation
IL-1β	Inflammation
Snail	EMT
ACTA2	Collagen synthesis (a-SMA)
CCND1	Proliferation/recruitment of fibroblasts
CCNB1	Proliferation/recruitment of fibroblasts
PLK1	Proliferation/recruitment of fibroblasts
Survivin	Proliferation/recruitment of fibroblasts
IL-6	Inflammation
IL-11	Inflammation
TNF-a	Inflammation
ZEB1	EMT
VEGF	EMT, fibroblasts
CTGF	Fibroblasts
ZO-1	Fibroblasts
MMP2	Fibroblasts
MMP9	Fibroblasts
CENP-A	Proliferation/recruitment of fibroblasts

## Data Availability

Data are contained within the article.

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
