# Peer review of "FOXM1: Functional Roles of FOXM1 in Non-Malignant Diseases"

_biomolecules, 2023, doi:10.3390/biom13050857_

Round 1

Reviewer 1 Report

General comments on FOXM1 review manuscript by Zhang et al.:

This is a nicely presented review on the roles of FOXM1 in malignant and non-malignant diseases based on the published articles till 2022. The review is appropriate and timely and would benefit the scientific community.

Role of FOXM1 in cancer is well established. However, its role in other human diseases has not been reviewed in detail. This review is valuable in this regard. The authors here briefly mention its role in cancer and describe further its additional roles in pulmonary fibrosis, diabetes, diseases of liver, renal, vascular, and heart, and rheumatoid arthritis with appropriate illustrations.

Some minor points to improve the manuscript:

1.     FOX/FOXM1 should be spelled out at its first appearance. It was spelled out later (Lines 50 and 110 of the pdf manuscript). Similarly, IPF, PAH (Line 420), AtoMs (Line 514) etc.

2.     Some abbreviated gene/protein names were described in many ways throughout the paper and these should be consistent. For example, FOXM1 (other abbreviations: FoxM1, Foxm1), Snail1 (SNAIL1), Wnt (WNT), Ace (ACE).

3.     Some figure legends, particularly Fig. 7 and 8 could be expanded.

4.     There are many typos in the manuscript due to the conjoined of words of 2 or 3. I suspect this may happen during the conversion of the word file to the pdf file. Examples of these conjoint word are partessentialelucidate (Line 40), hugesubstantialnomic (Line 42), noxitoxicironment (Line 47), veryerent bi-olbodilyctions (Line 55), muis tation (Line 80), anessential (Line 325), maysignificantly (Line 388).

The manuscript needs minor English editing and corrections of many typos.

There are many typos in the manuscript due to the conjoined of words of 2 or 3. I suspect this may happen during the conversion of the word file to the pdf file. Examples of these conjoint word are partessentialelucidate (Line 40), hugesubstantialnomic (Line 42), noxitoxicironment (Line 47), veryerent bi-olbodilyctions (Line 55), muis tation (Line 80), anessential (Line 325), maysignificantly (Line 388).

Author Response

Dear Reviewers 1.

We would like to thank the reviewers for their selfless dedication. We are pleased that the reviewers have given this very important opportunity for revision. We thank the reviewers for giving this manuscript a high rating of “This review is valuable in this regard. The authors here briefly mention its role in cancer and describe further its additional roles in pulmonary fibrosis, diabetes, diseases of liver, renal, vascular, and heart, and rheumatoid arthritis with appropriate illustrations". However, there are still some problems with this manuscript, and to address the reviewers' comments, we have carefully read and revised the manuscript line by line according to the reviewers' comments. In summary, we have carefully revised the manuscript in accordance with the reviewers' comments to address the reviewers' concerns. Thank you for your consideration of the revised manuscript. We hope that the revised manuscript will be considered suitable for publication in the journal biomolecules.

Sincerely.

Chao Liu, Professor

Hubei University of Science and Technology

Reviewer 2 Report

In this review paper, Zhang et al. collected information on the role of FOXM1 in benign diseases. The authors provide a large collection of knowledge on FOXM1. However, there are structural difficulties, such as when the authors provide a first detailed discussion in the introduction section before discussing the same topic in detail again later in the manuscript. Importantly, the authors’ references on FOXM1’s function in the cell cycle are largely outdated and to this point it is critical that the authors update their content to current models. In addition, the authors missed to highlight that most of the functions ascribed to FOXM1 are consequences from its pro-proliferative role by regulating the transcription of cell cycle genes .

Throughout the manuscript, the authors are using sweeping statements that are incorrect. For instance, “interaction with all kinds of transcriptional co-regulators” – the interactome is naturally limited and the authors need to be more precise about it. The authors must be more precise throughout the manuscript and correct sweeping statements.

Together, I cannot recommend publication of the manuscript in its current form, but I think that the authors will be able to improve their manuscript through a major revision.

More specifics

Lane 55: “veryerent” should be corrected.

Lane 75: While FOXM1 affects these processes, it directly regulates only few of them, e.g., proliferation, with the remaining effects being driven indirectly.

Lane 78/78: Instead of referring to another review article, the authors should point the authors to relevant original research. To this point, important studies on FOXM1 targets include PMIDs 22391450; 23109430; 23347430; 26100407; 27280975.

Lane 80: “muis tation” should be corrected.

Lane 91/92: It should be clarified that FOXM1 is itself regulated by the cell cycle. As a consequence, diseases that affect cell proliferation cause respective changes also in FOXM1 expression levels.

Lane 112/113: The references on FOXM1’s role in cell cycle progression are highly outdated. The authors need to update update their FOXM1-cell cycle part, e.g., through recent overview articles: PMID 35835684. For instance, data from the past decade showed that FOXM1 predominantly influences the G2/M transition and hardly the G1/S transition.

Section 2.2.: It should be pointed out that FOXM1 is critical for the cell cycle progression and thus proliferation of essentially all mammalian cells. Beta-cell proliferation and liver regeneration (section 3) are examples for FOXM1’s role in proliferation.

Figures 5 and 6: The illustrations suggest that FOXM1 is binding alone to DNA to regulate cell cycle/pro-proliferative genes. However, current models hold that FOXM1 binds together with the Myb-MuvB complex to regulate those genes (PMID 35835684).

Lane 410: The authors highlight that FOXM1 inhibits NF-kB signaling. It must be noted, however, that FOXM1 also has been shown to cooperate with NF-kB (PMID: 25159142).

Lane 556: While reference 114 shows that loss of FOXM1 can cause senescence, PMID 34728711 further strengthens this point as it shows that FOXM1 expression can bypass senescence.

Lane 624/625: The authors discuss a limited availability of drugs targeting FOXM1. I think the discussion would benefit from including recent research on peptides that contain FOXM1’s auto-inhibitory region (PMID 31244930).

Substantial English language editing is required. Syntax is a major issue throughout the manuscript.

Author Response

RE: FOXM1: Functional roles of FOXM1 in non-malignant diseases (Submission biomolecules-2364075)

Dear Reviewers.

We would like to thank the reviewers for their selfless dedication. We are very pleased that the reviewer has given us this very important opportunity to revise. We would like to thank the reviewers for suggesting many valuable changes to this article. Moreover, the reviewer provided several references on "Function and role of FOXM1 in cell cycle". These references play an important role in the revision of this manuscript. In addition, reviewers proposed that FOXM1 and MMB co-regulate downstream target genes, and then regulate cell cycle progression, which has certain value for our understanding of FOXM1's role in the regulation of pulmonary fibrosis, diabetes, liver, kidney, vascular and heart diseases, and rheumatoid arthritis. We would like to thank the reviewer again for their suggestions and opinions. In order to solve the reviewer's comments, we carefully read the manuscript line by line and modified it according to the reviewer's requirements. In summary, we have carefully revised the manuscript in accordance with the reviewer's comments to address the reviewer's concerns. Thank you for your consideration of the revised manuscript. We hope that the revised manuscript will be considered suitable for publication in the journal Biomolecules.

Sincerely.

Chao Liu, Professor

Hubei University of Science and Technology

Reviewer 3 Report

The manuscript review by Zhang and Li et al is an elaborate and illustrative description of the role of FOXM1 in non-malignant diseases. There are several reviews about its role in cancers however, this is unique as it describes in detail about the role of FOXM1 in non-malignant disease in particular. The review is well written with several schematic representations for better understanding of the subject. There are few major comments to be addressed for the improvement of the manuscript further.

Major comments

1.       Abstract: Since the authors have mentioned “This paper reviews the key roles and functions of FOXM1 in human cancer, pulmonary fibrosis, alcoholic steato-hepatitis, and diabetes to elucidate the role of FOXM1 in developing and progressing human diseases and makes suggestions for further research” and also mentioned about lung cancer and other cancers in the text, it would be interesting to add a para about the role of FOXM1 in all cancers as well. The authors can include separate heading such as “Role of FOXM1 in the progression of cancer” and mention all about different cancers very briefly with appropriate schema.

2.       Title: The title can be then changed to “FOXM1: Functional roles of FOXM1 in non-malignant and malignant diseases”.

3.       How does FOXM1 play role in immune responses to all the above diseases? The authors have mentioned few points, it would be interesting if the authors can mention in a separate para with appropriate heading.

4.       The authors are also suggested to include a table mentioning all the target genes and those genes regulated by FOXM1 (those depicted in schema) in each malignant and non-malignant diseases.

5.       The authors are suggested to make separate headings for non-malignant 1) lung diseases 2) Liver diseases 3) kidney diseases 4) Brain related 5) Cardio-related 6) Bone related 7) Reproductive system related 8) Skin related diseases and others such as Metabolic diseases and immune related diseases, and subheadings mentioning the name of disease. This way it is easy for the readers to focus on each type of diseases and the role of FOXM1 and any attempt of developing therapy related approaches for each of them. An alternate approach is to mention the organ related cancers (citing appropriate recent research articles) under each of these headings/subheadings instead of separate para (as mentioned in point #1).

6.       Additionally, add a concluding remark and future perspective for the review mentioning the lacuna and what is required to be made for highlighting the importance of FOXM1 in therapy point of view.

Minor Comments

Extensive English language and grammar editing is required from a professional editor.

Extensive English language and grammar editing is required from a professional editor.

Author Response

RE: FOXM1: Functional roles of FOXM1 in non-malignant diseases (Submission biomolecules-2364075)

Dear Reviewers.

We would like to thank the reviewers for their selfless dedication. We are pleased that the reviewers have given this very important opportunity for revision. We thank the reviewers for giving this manuscript a high rating of “This review is valuable in this regard. The authors here briefly mention its role in cancer and describe further its additional roles in pulmonary fibrosis, diabetes, diseases of liver, renal, vascular, and heart, and rheumatoid arthritis with appropriate illustrations". However, there are still some problems with this manuscript, and to address the reviewers' comments, we have carefully read and revised the manuscript line by line according to the reviewers' comments. In summary, we have carefully revised the manuscript in accordance with the reviewers' comments to address the reviewers' concerns. Thank you for your consideration of the revised manuscript. We hope that the revised manuscript will be considered suitable for publication in the journal biomolecules.

Sincerely.

Chao Liu, Professor

Hubei University of Science and Technology

Round 2

Reviewer 2 Report

The authors have addressed all my concerns and I recommend the manuscript for publication.